# Advances in the Application of Nanomaterials to the Treatment of Melanoma

**DOI:** 10.3390/pharmaceutics14102090

**Published:** 2022-09-30

**Authors:** Zeqi Wang, Yu Yu, Chuqiao Wang, Jin Li, Yan Pang

**Affiliations:** 1Department of Ophthalmology, Shanghai Ninth People’s Hospital, Shanghai Jiao Tong University School of Medicine, Shanghai 200011, China; 2Shanghai Key Laboratory of Orbital Diseases and Ocular Oncology, Shanghai 200011, China

**Keywords:** melanoma, nanoparticles, drug delivery

## Abstract

Melanoma can be divided into cutaneous melanoma, uveal melanoma, mucosal melanoma, etc. It is a very aggressive tumor that is prone to metastasis. Patients with metastatic melanoma have a poor prognosis and shorter survival. Although current melanoma treatments have been dramatically improved, there are still many problems such as systemic toxicity and the off-target effects of drugs. The use of nanoparticles may overcome some inadequacies of current melanoma treatments. In this review, we summarize the limitations of current therapies for cutaneous melanoma, uveal melanoma, and mucosal melanoma, as well as the adjunct role of nanoparticles in different treatment modalities. We suggest that nanomaterials may have an effective intervention in melanoma treatment in the future.

## 1. Introduction

Melanoma is a clinically common malignancy in various body tissues, such as the skin, iris, and mucosa. Its most common form is cutaneous. Melanoma is one of the fastest-growing malignancies with higher mortality and shorter survival time than others [1]. Although the incidence of melanoma varies by race and region, the incidence and mortality of melanoma have been increasing worldwide [2]. The incidence of this cancer in the Caucasian population has been steadily growing since the 1960s [1]. According to GLOBOCAN, in 2020, approximately 325,000 new cancer cases were diagnosed worldwide, of which 1.7% were diagnosed with cutaneous melanoma (CM), and 57,000 people died of melanoma [1]. Skin tumors are one of the most common cancers in the Caucasian population [3].

The WHO classification of skin tumors divides melanoma into two categories: etiologies related to sun exposure and etiologies not related to sun exposure, according to the mutational signatures, the anatomical site, and the epidemiology of melanoma [4]. Based on the degree of cumulative solar damage (CSD) histopathology in the surrounding skin, melanoma on sun-exposed skin was further classified into low CSD and high CSD. Low-CSD melanomas include superficial spreading melanomas, and high-CSD melanomas include lentigo malignant and desmoplastic melanomas. The “nonsolar” category includes acral melanoma (AM), mucosal melanoma (MM), and uveal melanoma (UM), among others [4]. In clinical work, melanoma is usually divided into different subtypes depending on the primary tumor site. These include CM, which occurs in the skin; AM, which arises from hairless skin on the palms, soles, and nail beds; UM, which develops from melanocytes in the uvea of the eye; and MM, the rarest subtype, which arises from melanocytes in the lining of internal tissue [5]. The different types of melanoma are race-specific. In the Caucasian population, CM is the primary type of the disease, while MM and AM represent only 1% to 7%; in Asia, the incidence of acral and MM is significantly higher than in the Caucasian population [6].

Currently, there are treatment options for melanoma, and the appropriate treatment options are selected based on the site and tumor stage. Surgical resection remains the first-choice therapeutic option, especially for CM [7]. For early-stage CM, excision with margins of healthy skin remains the best treatment option. Additionally, selective lymphadenectomy is required in the presence of regional lymphadenopathy with palpable regional lymph nodes [7]. Although surgery is the preferred treatment option, when a patient is diagnosed with distal metastasis or inoperable melanoma, systemic treatment methods, including immunotherapy, chemotherapy, and targeted therapy, must be implemented [8] (Figure 1).

In recent years, based on investigations into the molecular pathogenesis of melanoma and the immunology of tumors, there has been a revolution in the treatment of patients with advanced and unresectable melanoma, resulting in significant improvements in outcomes for patients [9]. The mortality for melanoma patients in the United States fell by 18% between 2013 and 2016 as new treatments for advanced patients became available [2].

Despite these innovative therapeutic options, however, the clinical management of advanced or metastatic melanoma remains difficult. Nanomaterials are considered capable of improving the limitations of existing treatment options and have positive significance in many aspects such as targeted drug delivery and the enhancement of therapeutic effects. Therefore, applying nanomaterials in melanoma is a promising therapeutic strategy to improve clinical outcomes. The application of nanomaterials in medicine offers unique opportunities to improve current therapies for cancer and other diseases. Nanomaterials can potentially enhance drug delivery efficiency and enable targeted delivery, limiting antineoplastic agents’ systemic toxic effects [10]. At present, a variety of nanomaterials have been extensively studied in tumor imaging, drug delivery, photothermal therapy, immunotherapy, et cetera, including quantum dots, liposomes, polymer nanoparticles, carbon nanotubes, metallic nanoparticles, and dendrimers [11] (Figure 2).

In this review, we primarily focus on the application of polymeric nanomaterials in different melanoma subtypes and treatment options and an overview of the recent development of the systemic treatment of melanoma. Summarizing recent attempts by scientists worldwide to use nanomaterials to treat melanoma treatments, we discuss how this information will drive the growth of novel and effective treatments of melanoma therapy for maximum effect.

## 2. Cutaneous Melanoma

CM is the most common type of melanoma, accounting for approximately 91.2% of melanoma cases, and less than 10% of melanoma occurs in extracutaneous sites [12]. CM is also one of the most aggressive forms of skin cancer, causing 55,500 deaths yearly [13]. The prognosis of CM is closely linked to early diagnosis and treatment. Once melanoma has spread, cancer can quickly become life-threatening. Although modern melanoma treatments target multiple mechanisms that enable cancer patients to develop and survive, CM still accounts for 0.6% of cancer deaths worldwide [1].

Melanoma has the highest mutation frequency of all cancer types, and its gene mutation frequency is greater than 100/Mb. In some cases, high mutation frequencies have been attributed to exposure to a wide range of well-known carcinogens, such as UV radiation in melanoma [14]. CM spreads to surrounding tissues and therefore is considered very aggressive and highly metastatic [15]. Metastatic melanoma is considered an advanced stage of skin cancer, which means malignant cells spread from their primary tumor to distant locations in the body. The early and effective treatment of melanoma is essential, as advanced stages are incurable and have a high mortality rate.

In recent years, with the increased understanding of the mechanism of melanoma, the treatment of melanoma has been continuously improved. Besides the surgical resection of primary lesions, current clinical treatments for CM include systemic chemotherapy, local radiotherapy, photodynamic therapy (PDT), and immunotherapy. Taking drug-targeted therapy, immunotherapy, and PDT as examples, each of these therapies has limitations. Thus, there is a need to refine the treatment of CM by designing biomaterials (such as molecules, atoms, and supramolecules) in the nanometric range (1–100 nm) [16].

### 2.1. Nanoparticles for Drug-Targeted Therapy

Chemotherapy plays a vital role in the comprehensive treatment of tumors. However, conventional chemotherapy suffers from several disadvantages, including poor solubility, and the excessive use of surfactants for the drug’s solubilization, resulting in systemic toxicities and adverse side effects [17]. In addition, the low drug concentration in tumors causes a population of drug-resistant cancer cells [18]. Nanotechnology has the potential to overcome these obstacles [11]. Nanoparticles (NPs) can be used to deliver drugs to corresponding targets in the body, thus significantly increasing the concentration of drugs in the target area, improving specificity and therapeutic effect, and reducing the potentially deleterious side effects. Nanoparticle-based drug delivery systems have succeeded tremendously in various tumor-targeted therapies [19,20]. In addition, multifunctional NPs can improve drug stability and control drug release, thereby reducing dosing frequency.

Drug delivery systems can be classified into three categories based on tumor tissue targeting: passive targeting, active targeting, and trigger-based drug delivery in response to internal/external stimuli. Passive targeting is based on the enhanced permeability and retention (EPR) effect, exploiting leaky vasculature and dysfunctional lymphatic systems in the tumorous zone to allow NPs loaded with drugs to accumulate [21]. Recent research shows that although NPs are enriched in tumor tissue through the spaces between tumor vascular endothelial cells (inter-endothelial gaps), these endothelial spaces are not responsible for the transport of NPs into solid tumors. At the same time, more than 97% of them use an active process through endothelial cells to enter cancers [22]. By modifying NPs, using receptors or molecules highly expressed in tumor tissues, active targeting achieves specific interactions between drug-loaded NPs and tumors through receptor-mediated endocytosis [23]. Additionally, by combining biologically responsive NPs with internal or external stimuli, such as pH, light, and acoustics, stimuli-triggered drug release is achieved. These stimuli-responsive NPs have not released the encapsulated therapeutic drugs unless in the local environment for corresponding stimuli, which minimizes the wastage of the drug before it reaches the tumor tissue [23].

#### 2.1.1. Passive Targeting

The EPR effect, first described by Matsumura and Maeda in 1986 [21], is a landmark for applying NPs in drug delivery. NPs preferentially accumulate at tumor sites due to the unique features of solid tumors (i.e., leaky vasculature and a dysfunctional lymphatic system). The passive targeting of NPs relies on the EPR effect. Currently, passive targeting has been used worldwide for NPs approved for diagnosis by clinical imaging and treatment [24], e.g., Doxil™ and Onivyde™ in the United States; Myocet™ and Onivyde™ in Europe; Mepact™, Genexol-PM™ from Korea; SMANCS™ from Japan; and many more [25].

From the oral, intravenous, or transdermal administration of NPs to the drugs they load into cancer cells, various physiological barriers in the body affect the efficiency of NP delivery (Figure 3) [26]. The mononuclear phagocytic system (MPS) and renal clearance can lead to the loss of NPs. NPs with a hydrodynamic diameter of less than 5.5 nm can be directly excreted by the kidneys [26]. The MPS also clears many NPs. As a cellular system with strong phagocytic capacity and defense functions in the body, the MPS is composed of macrophages, monocytes, and immature monocytes scattered in various body organs and tissues (liver, spleen, and other organs), and is capable of phagocytosing and absorbing NPs. Usually, NPs are cleared by Kupffer cells in the liver [27]. NPs escaping the mentioned biological barriers have a chance of reaching the tumor tissue. Once the NPs have reached the target cell, they still need to cross the cell membrane and escape the cell barrier to function. Endosomes are membranous organelles in animal cells that transport substances ingested by endocytosis to lysosomes. Once the nanocarrier is coated with membrane vesicles and enters the cell, it fuses with the first endosome. There is an ATP-driven proton pump on the endosome, which pumps hydrogen ions into the endosome to reduce the pH in the cavity from about 6.2 to about 5.5 [28]. This can promote the maturing of the first endosome and form the second endosome. The second endosome is fused with a lysosome containing a specific digestive enzyme, which results in the degradation of the drug or loaded nucleic acid [29]. Therefore, the ability of nanocarriers to escape from lysosomes is crucial for the delivery of their loaded drugs. Moreover, the therapeutic efficacy of the passive targeting of NPs is affected by the intratumoral heterogeneity of the effects of EPR. In tumors, the low perfusion (i.e., blood supply), the nonuniform distribution of blood flow and vascular permeability, and the dense microenvironment all lead to the uneven penetration and distribution of nanomedicines in tumors [30]. Permeability is not the only limiting factor. NPs are also hindered by the tumor interstitial matrix, which forms a barrier that does not allow them to penetrate deep into the tumor tissue [31]. Studies have confirmed that the particle size of an NP affects its penetration and delivery efficiency. For example, Tang et al. compared the biological characteristics of silica NPs of three different sizes (20, 50, and 200 nm) [32]. NPs of 50 nm exhibited the highest tumor penetration, slowest tumor clearance, and best efficacy against primary and metastatic tumors in vivo. Other physicochemical properties of NPs, including shape, elasticity, and surface charge, also affect interactions with physiological barriers and the tumor microenvironment [29], which is essential for optimizing NPs’ design and maximizing their functions. The shape of NPs is a crucial factor in determining their blood circulation and uptake by tumor cells and macrophages [33]. Tuning the elasticity of NPs may provide a means to improve the biological fate of NPs by improving circulation, reducing immune system uptake, and improving targeting [34]. Overall, the physicochemical properties of NPs can significantly affect their accumulation, retention, and penetration into tumors.

A meta-analysis of preclinical data based on tumor delivery platforms involving nanomaterials showed that approximately 0.7% of the injected dose (ID) of NPs reached the target tumor [26]. Although this percentage is small, it significantly improves the delivery efficiency of conventional chemotherapeutic drugs, considerably higher than most traditional chemotherapeutic drug formulations currently prevalent in the clinic. For example, a preclinical study by Vlerken et al. showed that the delivery efficiency of paclitaxel-loaded NPs was 0.6% ID compared to 0.2% ID for free paclitaxel [35], when the paclitaxel was encapsulated in the pH-responsive rapid releasing polymer, poly(beta-amino ester). Therefore, it is essential to build advanced nano-drug delivery systems to cross physical/biological barriers and increase the antitumor properties of chemical drugs. To address these issues, Hu et al. first coated sub-100-nm polymeric particles with bilayered RBC membranes, including both lipids and the corresponding surface proteins, in 2011 [36], which paved the way for later research of cell-membrane-covered NPs [37]. This novel biomimetic nano-drug delivery system can trick the host immune system, improve biocompatibility, and extend retention time [36]. As one of the different types of cell membranes, the red blood cell membrane (RBCM) is the most abundant and biocompatible and is known to penetrate biological barriers. Numerous studies have demonstrated the usefulness of the RBCM in encapsulating drug-loaded polymeric NPs [38]. In the latest study, Song et al. designed a novel polyethylene glycol (PEG)-FA NDDS (FRCS NP)-modified RBCM based on the refined “core-shell” structure of the RBCM [39]. The study found that FRCS NPs exhibited good biocompatibility and improved anti-melanoma efficacy in an acidic tumor microenvironment. Moreover, as the most abundant leukocytes in the systemic circulation, neutrophils can be recruited to sites of inflammation by the action of granulocyte colony-stimulating factors in the tumor inflammatory microenvironment. Zhou et al. reported a biomimetic drug delivery system provided by polylactic-glycolic acid (PLGA) NPs coated with a neutrophil membrane and targeted specificity for malignant melanoma [40]. NPs camouflaged in the cancer cell membrane have shown promising potential in treating tumors due to their homologous targeting ability, prolonged blood circulation, and immune escape. By encapsulating dacarbazine (DTIC) in the cavities of mesoporous silica NPs (MSNs) and coating their surfaces with cancer cell membrane fragments, the resulting DTIC@CMSNs exhibited efficient accumulation at tumor sites when injected intravenously into tumor-bearing mice [41].

Liposomes are the most common and well-studied nanocarriers for targeted drug delivery. As the first class of therapeutic NPs to be clinically approved for cancer treatment, they have attracted much attention due to their excellent biocompatibility, efficient delivery, and ability to protect therapeutic agents from degradation [42]. For example, Vecchione et al. proposed an oil-in-water nanoemulsion stabilized with lecithin and loaded with cobalt ferrite (Co_0.5_Fe_2.5_O_4_) nanocubes for photoacoustic and magnetic resonance imaging. A mouse melanoma xenograft model showed clear and significant time-dependent accumulation in tumor tissue in an in vivo photoacoustic study. Additionally, the curcumin-loaded nanoemulsion showed clear cytotoxicity to melanoma cancer cells, suggesting that the system has therapeutic conversion significance [43]. However, preventing drug leakage before tumor administration is a vital issue due to the instability of liposomes in vivo [44]. Another question is how to avoid the MPS in the blood. PEG conjugation of liposomes is the most common method used to modify liposomes, which reduces macrophage uptake and results in the prolonged presence of liposomes in the circulating blood [45]. However, this method does not significantly improve the stability of liposomes, and the introduction of PEG can cause severe allergic reactions. Therefore, Li et al. synthesized a new phosphocholine lipid-loading cytotoxic drug, Doxorubicin (Dox). This liposome exhibited increased blood circulation time and aggregated at the melanoma site in melanoma mice, showing higher stability, prolonged blood flow, and more precise release [45]. Vitamin C, ascorbic acid (AA), can affect melanocyte function by indirectly inhibiting tyrosinase activity by its antioxidant effect, thereby reducing melanin production and melanocyte synthesis [46]. In vitro studies on several human and murine cancer cell lines showed that AA at approximately 20 mM killed cancer cells without adversely affecting regular cell lines [47]. Additionally, Cha et al. showed that vitamin C supplementation significantly reduced B16FO melanoma metastasis in Gulo knockout mice [48]. However, due to the instability and hydrophilicity of AA in the blood, the bioavailability is low, and the dose required to achieve an effective concentration is high. Loading liposomes improved the bioavailability of AA. However, due to the high solubility of AA in water, the encapsulation of liposomes is complex. Therefore, several experiments performed chemical modification with AA by esterifying hydroxyl groups with palmitic acid to give palmitoyl ascorbate (PA). PA has higher encapsulation efficiency and can deliver PA to tumor tissue through liposomes, which may improve its ability to cross cell membranes while retaining the substantial antioxidant effect of parent AA. The practical therapeutic impact of this design has been validated in several tumor-bearing animal models, such as the murine model of melanoma lung metastases [49,50,51].

PEG and PLGA, mentioned above, are both United States Food and Drug Administration (FDA)-approved pharmaceutical excipients and are widely used in the pharmaceutical industry [52]. PEG-PLGA block copolymers have long been used to manufacture pegylated NPs to overcome protein adsorption and achieve prolonged circulation after systemic administration [53]. For example, in a recent study, Barjasteh et al. evaluated the use of MIL-100 (Fe) metal-organic frameworks with PEG coatings to load DTIC in the in vitro treatment of melanoma. Experimental results showed that the PEG coating changed the surface charge of the MOF from −2.8 ± 0.9 mV to −42.8 ± 1.2 mV, which improved the colloidal stability of the MOF and increased the time for drug release [54]. Moreover, methionine aminopeptidase 2 (MetAp2) is a well-recognized attractive molecular target for anti-angiogenic agents [55]. In a recent study, to reduce the systemic dose of a newly synthesized small-molecule inhibitor of MetAp2 while increasing drug solubility and targeting, Esa et al. prepared PLGA. These biodegradable polymers, as drug-loaded nanoplatforms, can effectively inhibit melanoma growth in mice [56]. Additionally, resveratrol (RSV) is a naturally occurring polyphenol with antioxidant, anti-inflammatory, anticancer, antibacterial, and anti-neurodegenerative effects [57]. Unfortunately, its low bioavailability, rapid hepatic metabolism, and in vivo degradation limit its physical benefits. To improve plasma stability and reduce the metabolic rate of RSV, Yee et al. achieved this by combining RSV with a low molecular weight mPEG-PLA copolymer, which showed a stable plasma stability profile and reduced hepatic metabolic rate [58].

MSNs have been widely studied as drug nanocarriers. With their mesoporous structure and large surface area, MSNs have significant advantages over conventional drug nanocarriers as drug delivery systems [59]. The carrier allows the on-demand release of high local concentrations of therapeutic drugs upon the application of specific stimuli and its ability to deliver the drug to particular locations in the body [60]. Indomethacin (IND) is a nonsteroidal anti-inflammatory drug derived from indoleacetic acid, which has adverse effects such as gastrointestinal discomfort. Ferreira et al. effectively inhibited tumor growth by incorporating IND into mesoporous silica nanoparticles, reducing the frequency of tumor tissue mitosis and inducing tumor cell apoptosis [61]. Additionally, Liu et al. loaded a drug-loaded viral nanoparticle (VLN) into MSNs and then encapsulated it with a lipid shell. This structure allows VLNs to remain stable during blood circulation. Upon reaching the tumor, VLNs release small molecule drugs in response to the microenvironment, synergistically modulating several cancer-related pathways [62].

The metal complexes of disulfiram derivatives and active metabolite copper diethyldithiocarbamate (CuET) are a recently reported promising candidate for cancer treatment. Its action is to prevent protein degradation by inhibiting the p97-NPL4 complex; it will eventually cause the accumulation of misfolded proteins and heat shock reactions, causing cancer cells to die [63]. Since cancer cells tend to accumulate misfolded proteins, leading to increased protein turnover [64], CuET can be used as a sensitizer in combination with current therapies for melanoma. However, clinical application is complex because CuET is insoluble in water and exhibits low bioavailability. The direct synthesis of nanoliposomal CuET (LP-CuET) by Paun et al., using ethanol injection, resulted in NPs exhibiting CuET encapsulation efficiency greater than 80%, and they remained colloidally stable in solution even after six months of storage, making them suitable for clinical use. LP-CuET was biocompatible and stable in plasma for several hours without any signs of hemolysis. Using nanoliposomal formulations in cancer cells has enhanced the cytotoxicity of LP-CuET, making CuET a potential adjuvant for currently approved therapies [65].

DTIC is one of the chemotherapeutic drugs for melanoma approved by the FDA. However, similar to other very potent antitumor drugs such as docetaxel (DTX), paclitaxel (PTX), and Dox, its clinical efficacy was suboptimal in patients with CM, which has an objective response rate of usually no more than 15% [23]. At the same time, poor solubility, poor stability, and severe toxicity to normal tissues further limit its clinical applications. Hyaluronic acid (HA), a natural polysaccharide that binds explicitly to overexpressed CD44 in tumor tissues [66], has been widely used as a tumor-targeting agent due to its high compatibility and biodegradability. Mishra and colleagues formulated HA-coated liposomes by solvent injection. They loaded two anti-melanoma drugs, DTIC and eugenol, which exhibited significant effects on cell migration and proliferation, and showed a significant inhibitory effect on melanoma. The liposomes showed 95.08% cytotoxicity at a DTIC concentration of 0.5 µg/mL, while the DTIC solution showed only 10.20% cytotoxicity at the same concentration [67].

Local anesthetics (LAs) are a group of small molecule drugs widely used in clinical anesthesia and analgesia. With the widespread involvement of LA in cancer-related surgery, there is growing clinical evidence that the use of LA during or after surgery generally improves patient outcomes [68]. Studies have shown that LAs can kill tumor cells if they come into direct contact with them [69]. Self-assembled peptide nanomaterials, as an emerging class of biomaterials with excellent biocompatibility and controllability, have shown great potential as advanced drug carriers [70]. Yang et al. designed a self-assembled GQY and prepared LA NPs using GQY as support material. Lipid nanoparticles significantly enhanced the cellular uptake of lidocaine and, in animal models, significantly inhibited the development and recurrence of solid tumors. The lipid nanoparticle system may be a promising bifunctional agent for pain relief and the inhibition of tumor recurrence [71].

In recent years, as the most commonly used metal NPs, AgNPs have received increasing attention, and their physicochemical properties, antibacterial activities, and toxicity to mammalian cells have been characterized [72]. Silver NPs (AgNPs) prepared and stabilized from various biologically active substances have a wide variety of biological and medical applications. Incorporating AgNPs with biologically active substances such as antioxidants can develop new systems with desired anticancer properties. Barbasz et al. used AgNPs prepared with various antioxidants such as gallic acid, (-)-epicatechin-3-gallate, and caffeine, which showed better effects on melanoma cells compared with AgNPs by themselves [73].

#### 2.1.2. Active Targeting

Active targeting, known as ligand-mediated targeting, is an approach that uses affinity ligands on the surface of NPs to increase specific retention and promote the aggregation and uptake of NPs in areas of tumors. The active targeting of NPs is proposed as a complementary strategy to passive targeting, increasing the relationship between NPs and cells by selecting appropriate ligands to bind to surface molecules or receptors that are overexpressed in organs, tissues, cells, or domains (Figure 4) [74]. Targeting specificity and delivery capacity are two essential aspects of assessing the effectiveness of the active targeting of NPs. Specificity depends on the biodistribution of NPs with ligand functionalization and how the ligand–NP conjugate system interacts with off-target molecules and cells, defined by the properties of the ligand and NPs. The transmission capacity is directly linked to the material and the structure of the NPs [74]. Actively targeted NPs must be close to their target antigen to recognize and interact with it [75] The incorporation of targeting ligands on the surface of NPs increases their cellular internalization by target cells, instead of increasing the number of NPs reaching tumor tissue [76]. Since the systemic clearance of NPs affects the amount available to the tumor, the increase in the affinity of NPs for targeted tumor antigens in active targeting is insufficient to compensate for the loss of natural clearance processes [74]. Meanwhile, since molecular targets are usually located in the extravascular space of tumors, NPs rely on the EPR effect to reach their targets [74]. Moreover, the active targeting of NPs faces additional physiological obstacles due to their interaction with the target cells. These factors explain why active targeting strategies did not fundamentally change NP delivery efficiency and why the blood circulation time of ligand-modified NPs must be optimized for optimal targeting. Taking the example of the chemotherapeutic drug DTIC, DTIC has limited efficacy in treating melanoma due to its hydrophobicity and short half-life [77]. In addition, DTIC is non-specifically toxic to normal cells, a common shortcoming of chemotherapeutic drugs. Carriers of passively targeted nanomaterials can reduce the harmful side effects of DTIC and improve its cytotoxicity to melanoma cells. However, they can effectively prolong blood circulation after adding modified nucleic acid aptamers to achieve active targeted therapy.

Except for immature erythroid cells, most normal cells have low transferrin receptor (TfR) expression, but actively proliferating tumor cells show high levels of TfR [78]. Therefore, using TfR as a tumor-targeting surface marker is a reasonable strategy. In 2009, Liu et al. used the transferrin–polyethyleneimine (Tf-PEI) delivery system to transport HIF-1α shRNA to distant tumors to silence HIF-1α in MM tissue [78]. Additionally, Filipczak and colleagues developed transferrin-targeting liposomes by targeting vesicles on cells that overexpress the transferrin receptor. The developed liposomes were triple-loaded with a mixture of ascorbic acid, mitoxantrone, and anacardic acid [79]. The study by Liu et al. showed that Tf-conjugated lipid–polymer hybrid nanoparticles loading plumbagin and administered intravenously resulted in a complete disappearance of 40% of B16-F10 melanoma tumors and the regression of 10% of the tumors without any signs of animal distress [80]. Yuan et al. prepared liposomes modified with Tf and cell-penetrating peptides and encapsulated two chemotherapeutic drugs, paclitaxel and Dox. Both in vivo and in vitro experiments have shown that the system can not only strongly target melanoma but also, at the same time, it can effectively penetrate melanoma cells [81].

In addition to transferrin receptors, folate receptors (FRs) are upregulated in many cancer cell types and have been widely used as targets for nanomedicine drug delivery. Chen et al. used biodegradable polymer nanogels to overcome multidrug resistance through folate targeting. The nanogels were modified with a folate–PEG conjugate by copper-free click chemistry to load the chemotherapeutic drug Dox covalently. Specific uptake has been shown in folate receptor-positive B16F10 melanoma cells [82]. Elechalawar et al. developed a novel FR-targeting ligand combining folic acid and cationic lipids. This combination of liposomes allowed significant tumor regression in a tumor model of melanoma in which FR is moderately expressed [83]. Song et al. prepared tumor-targeted NPs based on the conjugation of biomaterials, including sodium alginate, cholesterol, and folic acid. NPs accumulated precisely and efficiently in xenograft melanoma tumors. Transporting metformin Dox can trigger the pyroptosis, apoptosis, and necroptosis of melanoma cells, thereby blocking the progression of melanoma [83].

Cell-based carriers, or the addition of protein modifications associated with cells, can disguise drug-laden materials as “self” components during systemic circulation, helping them to evade clearance from the reticuloendothelial system, thereby enhancing medication use [84]. For example, Ukidve and colleagues designed an erythrocyte-based immune targeting system that delivers antigen NPs anchored on the surface of carrier cells to antigen-presenting cells (APCs) in the spleen [85]. Extracellular vesicles (EVs) are native nanoscale bilayer structures produced by almost all cell types, and they become significant mediators of intercellular communication [86]. Moreover, EVs released from tumor tissues exert systemic effects by promoting angiogenesis, metastasis, immunosuppression, and chemoresistance, which contribute to the progression of malignant tumors [87]. The proteomic signature of EVs may ensure the specific tumor-targeting potential of EVs while having the ability to be endocytosed by cancer cells more efficiently than liposomes [88]. Therefore, Patras and coworkers designed Dox-loaded, modified melanoma cell-derived electric vehicles for anti-melanoma therapy, exhibiting long systemic circulation times in mice. Furthermore, melanoma cells showed the preferential uptake of PEG-EV compared to conventional Dox-loaded liposomes [89]. Additionally, immunogenic outer membrane vesicles can effectively utilize immune cell phagocytosis and the inflammatory microenvironment of tumors to effectively target cancerous areas and reduce unnecessary drug release upon systemic administration. For example, Gao et al. used outer membrane vesicles from *E. coli* to encapsulate β-cyclodextrin-modified gold nanoparticles (GNPs) and adamantane-modified GNPs. Escherichia coli outer membrane vesicles can induce phagocytic uptake, followed by intracellular degradation releasing the contents, and then disperse GNPs without photothermal effects into GNPs with photothermal product aggregates. Due to inflammatory tropism [90], phagocytes transport intracellular GNP aggregates into tumor tissue in vivo and then complete targeted photothermal therapy (PTT) on the tumor [91]. In addition, He et al. used membranes derived from milk exosomes to encapsulate organosol D-peptides and Au3+ peptides to construct chiral Au peptide covalent infinite polymers as modified exosomes to develop technology for feasible nanostructure packaging. In this study, a transplanted malignant melanoma model demonstrated that a D-peptide-derived p53 activator could restore the p53 signaling pathway by oral administration and specifically degrade MDM2 killer cells in cancer cells [92].

Evidence suggests that phagocytes can still clear intravenously administered pegylated liposomes of the reticuloendothelial system (RES) [93]. Additionally, due to the characteristic structure of tumors, the ability of liposomal drugs to penetrate tumors is often poor. Therefore, to improve the therapeutic effect of liposome drugs, reduce the clearance rate of liposomes, and improve their ability to target and penetrate tumors, Altin et al. transplanted pCD47 (a His-tagged CD172a binding sequence) on liposomes. Binding to CD172a on phagocytes stimulates and triggers the signaling system in phagocytes and inhibits the phagocytic uptake of liposomes. After carrying lipophilic tracers, it was shown that modified liposomes could target tumor tissues and enhance drug penetration into tumors [94].

Once in the biological environment, NPs interact with biomolecules such as albumin in plasma and will form complex protein layers called a protein corona. These protein layers strongly influence the interaction of NPs with cells, particularly drug uptake. Based on the binding affinity and the time required for the exchange of adsorbed proteins, protein coronas can be divided into a high relationship “hard corona” that directly interacts with the surfaces of nanomaterials, requiring a time of long exchange and low-affinity adsorption, which can achieve biomolecules’ fast “soft corona” exchange [95]. The protein corona’s presence dramatically alters the nanocarriers’ surface properties and provides a novel biological property that affects their actual physical response in vivo. Based on this study, Meewan et al. showed that pegylated zein micelles could effectively reduce the uptake of macrophages and dendritic cells (DCs) in plasma without affecting the uptake of micelles by melanoma. This may be due to the presence of anti-opsonins such as albumin and apolipoprotein in the hard caps of proteins surrounding the micelles, which antagonize the biological effects of micelle-bound opsonins [96].

Enediyne natural products are commonly used for treating liver cancer and leukemia in the form of copolymers or antibody–drug conjugates. Tiancimycin A is a new anthraquinone-fused enediyne that wholly, rapidly, and completely kills tumor cells. Feng et al. encapsulated Tiancimycin A in a functionalized liposome and demonstrated its antitumor activity using mice with melanoma. These NPs showed potent cytotoxicity against several tumor cells within 8 h. These NPs reduced tumor metastasis in a metastasis model, which demonstrates a better transformation [97].

Among the various NPs under investigation, protein NPs, such as human serum albumin (HSA) NPs, are emerging as promising candidates for their biocompatibility, non-immunogenicity, and tumor-targeting capabilities. The first HSA nanoparticles, albumin-bound paclitaxel (Abraxane^®^), have entered the clinic in advanced pancreatic and breast cancers [98]. Kumbham et al. used OA (a natural pentacyclic triterpene acid with anti-inflammatory and antitumor effects) and HSA to construct NPs carrying Dox. Dox–OA has the advantage of dual drug delivery with synergistic interaction [18].

Since most CM has activating mutations in the proto-oncogene BRAF, which is part of the Ras-Raf-MeCMk-Erk signaling pathway, this provides an essential direction for targeted therapy in melanoma targeting the MAPK pathway (BRAF and MEK inhibitors) [99]. Inhibitors, including Vemurafenib, have shown clinical benefits in the treatment of patients with melanoma [99]. However, the therapeutic effect is limited due to the need for high doses, the poor bioavailability of the drug, adaptive drug resistance behavior, and the reactivation of the MAPK pathway from drug-induced stress responses after repeated treatment cycles [100]. To improve this, as early as 2008, Tran et al. developed a nanoliposomal-ultrasound-mediated method using cationic nanoliposomes for delivering small interfering RNA targeting BRAF and Akt3 into melanocytic tumors present in the skin to retard melanoma development [101]. Imlimthan et al. designed vemurafenib-loaded nanocrystalline cellulose nanoparticles labeled with the radionuclide Lutetium-177, which combined radiotherapy and targeted therapy with NPs. The ideal treatment effect was obtained in mice with metastatic lung melanoma [100]. Mi and colleagues designed a porous silicon-based micro/nanocomposite that allows the simultaneous delivery of chemotherapeutic drugs and small interfering RNAs into the lungs after intravenous injection. The pores of the silica microparticles were loaded with liposomes containing BRAF siRNA, while the surface was bound with polymer nanoparticles encapsulating docetaxel. The synergistic antitumor effect of NPs has been confirmed in in vitro and in vivo experiments [102]. To better penetrate the epidermis of the skin, Tham et al. synthesized a mesoporous nanovehicle to co-load photosensitizers and MAPK inhibitors with microneedle technology [103]. Besides, for the treatment of vemurafenib-resistant melanoma, Fu et al. developed a nanoliposome loaded with protein kinase C inhibitor, palmitoyl-dl-carnitine chloride (PC), and BRD4-targeting PROTAC as a novel approach to Inhibit angiogenesis in BRAF-mutated malignant melanoma [104].

#### 2.1.3. Stimuli-Responsive Nanomaterials

Another obstacle limiting the therapeutic efficacy of NPs is the premature and non-specific release of encapsulated medicinal drugs. The development of NPs capable of responding to specific microenvironmental stimuli to release therapeutic agents “on demand” in a spatiotemporally controlled manner is an exciting solution. The first reports of stimuli-responsive NPs for cancer treatment appeared in the late 1970s [105], describing temperature-responsive liposomes for enhanced local drug release by hyperthermia. The design of stimuli-responsive NPs involves biocompatible materials that can undergo hydrolysis, cleavage, protonation, or conformational changes in response to intrinsic or extrinsic stimuli (Figure 5) [23,29]. Intrinsic stimuli include pH, enzymes, reducing agents, etc., and extrinsic stimuli include heat, light, electric fields, ultrasound, and magnetic fields. In contrast, extrinsic stimuli are more accessible to control precisely than intrinsic stimuli. Most stimulus-responsive NPs that have entered clinical trials and been approved are based on triggering external stimuli [106] (e.g., ThermoDox, NanoTherm, and MTC-Dox). In the latest study, Di-Cicco et al. report the preparation and characterization of a natural light-sensitive nanocarrier. It had shown promising efficacy against melanoma cells in vitro and in melanoma xenograft mouse models [107]. Moreover, Yuan et al. proposed and synthesized a novel doping cascade biocatalyst as a peroxisome mimetic, consisting of several enzyme mimetics, such as glucose oxidase mimics (Au nanoparticles for producing H_2_O_2_) and heme-mimetic atomic catalytic centers (Fe-porphyrin for ROS generation). The synthesized doping cascade biocatalysts suggested that the therapeutic effect of malignant melanoma can be significantly enhanced in in vivo animal data [108].

Although condition-responsive NPs can effectively improve drug conversion rates, issues such as material biocompatibility and stimuli compliance still require further study. The endogenous stimuli represented by the pH response have a good meaning. Shear-thinning biomaterials (STBs) have unique mechanical properties that reduce their viscosity under shear stress and restore them to their initial value after injection. Compared to conventional chemotherapy in systemic circulation, STBs allow the targeted and sustained delivery of chemotherapeutic drugs while minimizing off-target effects [109]. Since the microenvironment of malignant tumors is acidic (~pH 5.6–6.8) [110], using pH-sensitive biomaterials for drug delivery is a good option. For example, LAPONITE^®^ is a synthetic clay mineral with a structure similar to natural hectorite [111]. Previous studies have shown that LAPONITE^®^ achieves pH-dependent Dox release on target tumors [112]. A recent survey of gelatin/LAPONITE^®^ based on injectable STBs by Lee et al. revealed that the pH responsiveness and rheological properties of melanoma-targeting STBs could be regulated by controlling the composition ratio of gelatin/LAPONITE^®^, which reflects the therapeutic properties of STBs. Drugs can be delivered by personalized, highly controllable, and sustained-release mechanisms [109]. Pyrimidine analog 5-fluorouracil (5-FU) is a potent chemotherapeutic agent widely used to treat skin cancer. Although it is effective in treating malignant tumors, due to its rapid metabolism, low bioavailability, and short half-life, as well as its high bioavailability in other tissues by intravenous infusion as a hydrophilic drug, it has serious adverse effects, and the systemic dosage of 5-FU is restricted. Meanwhile, since melanoma is located between the dermis and epidermis, high molecular weight and hydrophilic biomolecules have poor penetration into the skin’s stratum corneum—5-FU needs a desired, effective method to deliver therapeutic drugs to the site. To address this issue, Pourmanouchehri et al. designed a pH-sensitive micellar hydrogel system based on deoxycholic acid micelles and carboxymethyl chitosanhydrogels to improve 5-FU efficacy against skin cancer and reduce its systemic side effects by enhancing its delivery to the skin [113].

### 2.2. Nanoparticles for Immunotherapy

Cancer cells constantly adapt to their host’s defenses by manipulating intrinsic and extrinsic biological pathways. In recent years, based on the concept of enhancing the activation of the endogenous immune system against cancer cells, immunotherapy has emerged due to its powerful effects in eradicating tumors and preventing recurrence [114]. Unlike traditional therapies such as surgery, chemotherapy, or radiotherapy, cancer immunotherapy has the potential to activate immune responses against tumor cells and improve the immunogenicity of the tumor microenvironment by triggering a durable immune memory to protect patients against metastasis [115]. The joint discovery of James P. Allison and Tasuku Honjo, winner of the 2018 Nobel Prize in Physiology or Medicine, treats cancer by inhibiting the downregulation of immune checkpoints [116,117].

The first immune checkpoint inhibitors, namely antibodies specifically targeting cytotoxic immunoregulatory molecules T lymphocyte-associated protein four and programmed cell death protein 1, were approved by the FDA in 2011 and 2014 for the treatment of unresectable or metastatic melanoma [118,119]. Like chemotherapy, most immunotherapies are applied systemically and aim to generate a systemic antitumor immune response. While current immunotherapies often fail to eliminate cancer due to local tumor-mediated immunosuppression, polymeric nanomaterials can effectively change the tumor microenvironment from immunosuppression to immune activation and stimulate antigen presentation through different pathways (Figure 6), which effectively promotes the therapeutic effect of CM [120].

#### 2.2.1. Immune Checkpoint Blockade

Immune checkpoint blockade therapy, targeting the well-known PD-1 and its ligands, has shown promising clinical results in the various malignancies represented by melanoma [121,122,123,124]. Anti-PD-1 monotherapy has been consolidated as one of the first-line therapies for advanced melanoma. However, a considerable proportion of patients with advanced melanoma remain unresponsive to anti-PD-1 monotherapy [125]. In recent studies related to melanoma, small interfering RNAs or antibodies can block the PD-1/PD-L1 pathway, but the effect of this blockade is temporary and reversible [126]. Therefore, using nanomaterials to achieve the precise and permanent inhibition of PD-L1 gene expression in melanoma cells has become a new challenge for immunotherapy. Thus, Wei et al. successfully constructed a PD-L1 gene-editing system by delivering a PD-L1 knockout plasmid mediated by the dendrimer derivative AP-PAMAM, thereby achieving the immune killing of tumor cells. In the B16F10 mouse melanoma cell line model, the alteration of PD-L1 expression levels and the antitumor efficacy induced by PD-L1 knockdown has been validated [127]. Similarly, Kwak et al. combined siPD-L1 with a polymer carrier composed of disulfide cross-linked polyethyleneimine and dermatan sulfate. The siPD-L1/pd complex entered B16F10 melanoma cells in a cell-specific manner and inhibited the expression of PD-L1 and melanoma-specific genes. It showed an excellent anti-melanoma effect in immunocompetent C57BL/6 mice and immunocompromised Balb/c nude mice [128].

#### 2.2.2. Adoptive T Cell Therapy

Adoptive cell therapy is a highly personalized cancer therapy that induces complete and long-lasting regression in melanoma patients by targeting cancer-unique somatic mutations through natural tumor-reactive lymphocytes [129]. Adoptive cell therapy typically relies on autologous APCs to deliver antigens to promote the expansion of CD8 effector T cells. DCs are a group of specialized APCs that play a central role in adaptive immune responses by capturing tumor-specific antigens for delivery to T cells [130]. However, solid tumors often secrete immunosuppressive factors, and surface antigens specific to aggressive tumor cells can quickly mutate or even disappear, which renders DC unable to recognize malignant cells and leads to tumor immune escape [131]. The targeted induction of immunogenic cell death (ICD) can turn tumors into vaccines in situ, thereby enhancing the ability of DCs to capture antigens. Evidence suggests that mitochondria and endoplasmic reticulum are two critical subcellular structures in the ICD process [132,133]. Based on this, Liu et al. engineered bidirectional damage to mitochondria and endoplasmic reticulum in tumor cells by nanoplatforms, through the two pathways that work together to promote DC activation. Studies have demonstrated that this strategy can stimulate a robust immune response to suppress primary melanoma and prevent postoperative melanoma tumor recurrence [133]. Stimulators of interferon genes (STINGs), such as cyclic dinucleotides (CDNs), stimulate dendritic cell maturation and the cross-presentation of tumor antigens for subsequent T-cell priming [134] via intratumoral injection to obtain an effective anti-cancer treatment [135]. Accordingly, Zheng et al. reported ADU-S100, a CDN STING agonist, formulated with chimeric polymers to significantly improve tumor retention and cytosolic delivery, promoting the activation of the pathway STING and the antitumor immunotherapy of malignant melanoma in mice [136]. In addition, studies have shown that the protein YTHDF1 is involved in DC-related antitumor immune processes. Its downregulation can effectively improve the tumor antigen cross-presentation ability of DCs, which is conducive to the subsequent infiltration of cytotoxic T cells into tumor cells in the tumor area and enhances antitumor immunity [137]. Accordingly, Ouyang et al. constructed a zwitterion-mannose-modified Au DENP nanoplatform to target YTHDF1 gene-silencing DCs. In a mouse model of melanoma, the combination of this nano-delivery platform with PD-L1 was effective against immune-resistant tumors, tumor metastasis, and tumor recurrence [138]. Additionally, while DCs are composed of potent APCs that can be used to expand T cells for adoptive transfer and have shown some clinical success, expansion using DCs takes weeks or months to generate large numbers of tumor-specific cells. Therefore, Ichikawa et al. used a novel artificial nanoparticle-based paramagnetic antigen-presenting cell that combines anti-CD28 co-stimulatory and antigen peptide-loaded human MHC class I molecules to develop a large number of antigen-specific T cells in melanoma patients within 14 days [139].

Besides stimulating APCs, nanomaterials can also stimulate T-cell activity through other pathways, enhance T-cell tumor infiltration, and kill melanoma. For example, Song et al. designed dynamic intracellular acid-activatable NPs that trigger immunogenicity by inducing ferroptosis in tumor cells. The combination of nanoparticle-induced ferroptosis and programmed death ligand one blockade effectively inhibited the growth of B16-F10 melanoma [140]. The synergistic effect of metallic nanomaterials and immunotherapy benefits the immune microenvironment and produces potent anti-cancer therapeutic effects with minimal toxicity. The study by Kuang et al. showed that S-AgNPs exhibited excellent local antitumor activity and mild systemic immunotoxicity by stimulating the infiltration of activated CD8+ T cells and the upregulation of PD-L1 in the tumor region in both in vivo and in vitro experiments [141]. Based on previous studies [124], the results suggest that S-AgNPs could be potential adjuvants for immunotherapy [141]. Furthermore, Liu et al. engineered a spontaneous co-assembly between three metal-organic coordination polymers to construct tumor-targeted metal-organic NPs bearing the Wnt inhibitor carnosic acid. The nano-delivery system exhibits preferential accumulation targeting the tumor microenvironment and enhancing intratumoral T-cell infiltration, thereby enhancing immunotherapy in an allograft mouse model role of B16F10 melanoma [142].

#### 2.2.3. Cytokine-Based Drugs

Cytokines are essential immunotherapy drugs approved for the treatment of human cancers. For example, IL-2 was approved for treating metastatic renal cell carcinoma and advanced melanoma in 1992 and 1998 [143]. However, the systemic administration of cytokines often fails to achieve sufficient concentrations of immune cells in tumors due to the dose-limiting toxicity. For this reason, Liu et al. designed a new lipid nanoparticle-mediated mRNA formulation of several cytokines that can induce intense tumor infiltration by immune effectors and inhibit tumor growth while reducing toxicity [144]. However, at the same time, cytokine release syndrome is the main side effect of immunotherapy. This syndrome occurs when immune cells are systemically overactivated and release large amounts of cytokines such as TNF-α, IFN-γ, and IL-6 [145]. Moreover, the combination of kinase and immune checkpoint inhibitors induces an excessive immune response in patients with metastatic melanoma [146]. Therefore, when nanomaterials are involved in melanoma immunotherapy, cytokine release syndrome immunotherapy should be minimized to avoid systemic overactivation. Recombinant human interleukin-2 (rIL-2) has been considered a potent immune activator in cancer immunotherapy. Due to the rapid clearance of intravenous rIL-2, patients receive frequent injections, resulting in increased systemic toxicity, varying concentrations, and significant systemic side effects associated with rIL-2, such as vascular leak syndrome and pulmonary edema [147]. To address this issue, Kim et al. designed a peritumoral injection formulation of rIL-2, which contains unmodified rIL-2 and silica NPs named DegradaBALL. The local injection of recombinant IL-2 can form rIL-2 depots in melanoma, effectively induce a local immune response, reduce the excessive activation of systemic immune cells, have a highly enhanced therapeutic effect, and significantly reduce the impact of rIL-2 on systemic adverse effects [148].

#### 2.2.4. Cancer Vaccines

The methods of cancer vaccines involve the use of tumor antigens to trigger an immune response. Nano-vaccines have the following advantages: they protect antigens from degradation, improve vaccine delivery efficiency, and enhance the cross-presentation of tumor antigens [149]. In a recent study, Chen et al. fabricated a nano vaccine with DC-targeting capability. The vaccine was able to co-encapsulate with antigen and adjuvant via electrostatic interactions. This positively charged nano vaccine can promote the endocytosis, maturation, and cross-presentation of DCs [149]. Nguyen et al. designed an injectable dual-scale mesoporous silica vaccine composed of mesoporous silica microrods (MSRs) and MSNs. This system enhanced and prolonged the cellular uptake of the MSN vaccine by recruiting large numbers of DCs into the MSR microporous scaffolds [150]. Berti et al. showed that poly(lactic-co-glycolic acid) (PLGA) NPs could improve the supply of tumor lysate to DCs [151]. A significant obstacle to the vaccine is the inability of DCs to effectively present foreign antigens to cytotoxic CD8 T cells. Therefore, in Lee et al.’s study, the liposome was channel-modified and loaded with polyethyleneimine as a nano-vaccine, which increases the liposome’s pH-responsiveness to facilitate the direct delivery of antigens to the cytoplasm of APCs [152]. Additionally, Song et al. developed a toll-like 7/8 agonist–epitope conjugate (TLR7/8a epitope) as a self-assembled and carrier-free nano-vaccine platform to efficiently introduce antigens and adjuvants. The TLR7/8a epitope nano-vaccines can prolong local retention time and improve lymph node drainage efficiency, causing significantly higher CD8 T-cell immunity levels than conventional vaccine formulations [153].

Transcutaneous immunization has the advantages of safety, high efficiency, non-invasion, and convenience of use. The key to the transcutaneous immunization system is the targeted transdermal delivery of antigens to DCs. In the study by Song et al., a transdermal tumor vaccine delivery system showed good performance in targeting DCs. It was observed that the vector loaded with antigen and adjuvant could effectively induce DC maturation in vitro [154].

In situ vaccination (ISV) can produce systemic antitumor immunity to metastatic cancer diseases [120]. Conventional vaccines contain antigens and immune agents. The tumor itself provides the antigen for ISV, and the treatment only uses the immune agent directly on the tumor. ISV is used for cancer immunotherapy, small-molecule proteins, or viral tumors to activate the pathogen recognition receptors that alter the tumor microenvironment and cause systemic antitumor immunity [155]. Various oncolytic viruses with ISV functions have been approved for treatment or are undergoing clinical trials, such as PVSRIPO, an engineered oncolytic poliovirus [156]. In recent years, ISV has also been extensively studied in CM. For example, Koudelka et al. studied the role of the in situ inoculation of the plant virus Cowpea Mosaic Virus (CPMV) against melanoma. CPMV has been shown to bind specifically to the mammalian protein vimentin, a cytoskeletal protein involved in various cellular functions that can be surface-expressed or secreted in tumor cells, endothelial cells, and immune cells [157]. Studies have confirmed that the CPMV-ISV manages the IFNβ signaling pathway by activating the TLR7 in the mouse model of skin melanoma, showing a good antitumor effect [155]. Jiang et al. researched the nano PC7A vaccine and confirmed that intratumoral delivery had stronger antitumor immunity and curative effects [158].

The therapeutic mRNA vaccine has recently developed rapidly [159] The mRNA vaccine can code one or more certain tumor antigens thanks to a precise sequence design. Furthermore, after translating intracellular proteins and treating the antigen, it is finally transmitted to T cells by DCs [160]. Recently, the search for therapeutic mRNA in melanoma has progressed very well. For example, Li et al. converted therapeutic miRNA into an infinite auric-sulfhydryl coordination supramolecular miRNA, termed IacsRNA, with near-spherical nanostructure, high colloid as anti-hydrolysis stability, and low macrophage uptake [161]. Li et al. used bacteria-derived outer membrane vesicles as an mRNA delivery platform by genetically modifying the surface modification of RNA-binding protein L7Ae and lysosomal escape protein listeriolysin O, which can significantly inhibit the progression of melanoma [162].

### 2.3. Photothermal and Photodynamic Therapy

The ultimate goal of nanoparticle-based phototherapy, including PTT, which generates heat, and PDT, which produces reactive oxygen species (ROS) and induces various antitumor immunities, is to kill tumors. In recent years, PDT and PTT have been proposed as new therapeutic options for tumor ablation. PTT and PDT are composed mainly of near-infrared (NIR) light and NPs, corresponding to photosensitizers (PS) and photothermal agents (PTA). The application of PTT is based on activating PTA at specific wavelengths of light to kill cancer cells. The antitumor effect of PDT is to generate single ROS and free radicals with cytotoxicity, short half-life, and low diffusivity, leading to apoptosis, autophagy, and tumor cell necrosis [163]. Different effects can be selectively controlled by controlling the intensity of the light source [164].

#### 2.3.1. Photothermal Therapy

In detail, based on the PTA, PTT can convert NIR light energy into heat to kill tumor cells [165]. Unlike UV and visible light, NIR light causes much less damage after penetrating deep tissue [166]. Although it has the advantages of spatio-temporal controllability, high specificity, and common side effects, the progressive thermotolerance of tumor cells hampers the therapeutic development [167], and the non-specific targeting of PTA leads to insufficient concentration at the tumor site and systemic toxicity [168]. Further research is still needed to refine the therapeutic role of PTT in melanoma. Fe_3_O_4_ is a nanomaterial that has received much attention and has been approved by the FDA as a safe biomaterial without long-term toxicity [169]. The exposure of iron oxide NPs to NIR light facilitates NIR-induced hyperthermia [170]. Wang and co-workers investigated the role of Fe_3_O_4_ nanoparticle clusters in melanoma PTT treatment. In in vivo studies in melanoma, model mice demonstrated the intratumoral delivery of nanoparticle clusters, and NIR light significantly inhibited the growth of implanted tumor xenografts [171]. Moreover, according to the principle that histone deacetylase (HDACis) inhibitors can inhibit HDAC activity, activate the expression of antitumor genes, and accelerate tumor cell differentiation and apoptosis [172], Wang et al. used Fe_3_O_4_ NPs. A chidamide-loaded magnetic polypyrrole nanocomposite was designed as a visualized photothermal cancer chemotherapy to combat tumor heat resistance and metastasis. It showed sound anticancer effects in in vivo and in vitro melanoma experiments and inhibited melanoma metastasis in mice [173].

#### 2.3.2. Photodynamic Therapy

PDT is a modern non-invasive therapy widely used to treat oncological and non-neoplastic diseases [174]. PDT is based on the uptake of PS by tumor cells and their excitation with appropriate wavelengths of light to induce tumor damage due to the production of ROS cytotoxicity [15,175]. Specifically, there are three main ways to produce antitumor effects: (1) direct destructive effect on tumor cells; (2) generating ROS, causing irreversible damage to cells and microvessels; and (3) dead cells stimulating ICD [163]. Since PS tends to bind preferentially to low-density lipoproteins (LDL), and vigorously dividing cancer cells show increased uptake of LDL lipoproteins, LDL may act as a “transporter” for PS to cancerous tissue. This biodistribution characteristic allows the accumulation of photosensitizers in cancer cells at significantly higher concentrations than in normal cells. Thus, PDT has several advantages over traditional cancer therapies, including low invasiveness, precise tumor targeting capability, low morbidity, and is well tolerated by patients [15].

Most PS are highly hydrophobic substances that aggregate in aqueous environments, and PS must remain in the monomeric form to be photoactive, so increasing the hydrophilic properties of PS can enhance the efficacy of PDT [174]. PS can be encapsulated or immobilized on nanoplatforms by covalent and non-covalent interactions [176], which effectively improve the bioavailability of hydrophobic PS by forming covalent bonds with hydrophilic polymer molecules [177]. Meanwhile, using polymeric NPs (e.g., micelles) in PDT can target the delivery of more photosensitizer molecules to the tumor area and prevent photosensitizer degradation before reaching the target tumor tissue [178]. For example, in the latest study, Trindade et al. designed a gelatin nanoparticle loaded with aluminum chloride phthalocyanine for PDT. In vitro experiments showed that the nanoparticle anti-cancer pellet preparation effect on B16-F10 murine melanoma cells is better than that of free-chloride phthalocyanine [179]. In addition, the use of polymers allows the simultaneous attachment of other ligands to PS molecules, such as contrast substances or fluorescent markers, thereby allowing the exploration of clinical images [174,180].

The use of NPs may also allow targeted approaches to specific receptors, thereby increasing the selectivity of PDT. For example, nanomedicines with negative surface charges typically circulate in the blood for long periods; however, negative charges also inhibit cellular uptake by tumor cells and reduce intracellular drug concentrations. Therefore, nano drugs with charge reversal capability (from negative to positive control) are needed to potentially enable long-term circulation and enhanced cellular uptake [181]. As a result, Huang et al. designed a charge-reversal nano-delivery system to improve targeted accumulation in systemic circulation and tumor tissue and enhance intracellular uptake. In this design, self-assembled NPs (PPC) were synthesized and then increased HA coating by electronic interaction to obtain negatively charged PPC@HA. Upon reaching the tumor environment, the outermost coating of HA was degraded to release free PPC. The positive charge on the surface of the PPC enhanced cellular uptake by B16 cells. Intracellular PPC is then broken down to release more drugs, allowing laser irradiation for combined chemo-photodynamic therapy (Figure 7) [181]. Another major limitation of PDT is its low depth of light penetration through pigmented lesions [182]. The limited penetration depth of external excitation light can significantly affect the therapeutic effect of PDT and its clinical application. Here, Yang et al. engineered bioluminescent bacteria by transforming an attenuated strain of Salmonella typhimurium so that the firefly luciferase expression plasmid is used as the internal light source to evenly illuminate the entire tumor, so as to inhibit the growth of melanoma [183].

The combination of MSNs and the second-generation photosensitizer Verteporfin has been further investigated. Combining Verteporfin with MSN, Clemente et al. engineered Ver-MSN as an effective nanoplatform to enhance cell-loading capacity and uptake. In vitro and in vivo studies have shown that the stimulation of Ver-MSNs under red light (693 nm) inhibits melanoma growth [184]. In a study by Rizzi et al., Ver-MSN-based PDT did not affect cell proliferation in a normal human keratinocyte line (HaCaT) or a low metastasis melanoma cell line (A375P) but could affect the highly invasive SK-MEL-28 melanoma cell, showing better therapeutic effects on highly aggressive melanomas [185]. Additionally, the study by Clemente et al. showed that the local administration of Ver-MSN PDT reduced melanoma lymphangiogenesis and lung micrometastases [186].

#### 2.3.3. Combination Therapy

In the latest research, using nanomaterials to achieve the combined treatment of PDT/PTT with chemotherapy or immunotherapy has good prospects. Lai et al. designed a light-triggered sequential release liposome containing free sunitinib and Dox small-sized polymer nanoparticles, Dox-NPs. The liposomal membrane was spiked with the photosensitizer and hybridized with RBCM to impart biomimetic characteristics. NIR light-induced membrane permeabilization triggers the “ultra-rapid” and “complete” release of sunitinib (100% release in 5 min) for the reversal of the immunosuppressive tumor environment [187]. Additionally, Xie et al. designed hollow porous gold nanocages (MLI-AuNCs) as carriers to deliver monophosphoryl lipid A, an immunogen for aggressive melanoma, and photosensitizing indocyanine green. MLI-AuNCs under NIR irradiation destroyed the primary tumor through the PTT/PDT effect of AuNCs and indocyanine green. They elicited a potent antitumor immune response through melanoma-associated antigens and monophosphoryl lipid A released in situ, showing a significant antitumor result [188].

The treatment of melanoma requires not only the removal of cancerous cells from the skin but also the regeneration of the skin to heal the defect. To this end, Shan et al. developed a two-layer microneedle platform with a dual function of the synergistic treatment of chemo-photothermal melanoma and skin regeneration. Microneedling is well known for its application in macromolecular drug delivery for its high safety, painless invasion, and simple administration [189]. Shan et al. selected HA to prepare soluble microneedles, which can be quickly dissolved after piercing the skin, releasing a new type of indocyanine green and curcumin NPs carried inside for the chemical combined PTT of melanoma. Meanwhile, a support layer composed of sodium alginate/gelatin/HA was left to cover the wound and promote the proliferation of endothelial cells and fibroblasts to enhance skin regeneration [182]. Additionally, Sutrisno et al. designed a bifunctional composite scaffold, which had an excellent photothermal effect and could promote fibroblast proliferation and the extracellular matrix component gene expression of skin tissue regeneration [190]. Wei and co-workers have developed a novel protein-based bifunctional photothermal bioadhesive for non-invasive tumor treatment and skin tissue regeneration [191].

## 3. Uveal Melanoma

UM is the second most common melanoma after CM and the most common intraocular malignancy in adults. The most common site of UM is the choroid. Shields et al. studied 8033 patients with UM [192] and found that 4% of tumors were located in the iris, 6% in the ciliary body, and 90% in the choroid. UM is more common in the elderly, with a gradual increase in incidence with age [193,194]. UM is characterized by an extremely high capacity to metastasize [195]. The uvea is one of the most capillary-rich tissues in the body but lacks lymphatic drainage. Therefore, the metastatic spread of UM occurs strictly hematogenously [196]. The liver is the most common site of metastasis in UM, but tumor metastasis can also occur in other organs, such as the lungs, lymph nodes, bones, skin, and brain. Therefore, the prompt and effective treatment of UM is of great importance. Currently, the clinical treatment of UM mainly includes transpupillary thermotherapy PDT, iodine-125 radiation therapy for small and medium ciliary melanoma, proton beam radiotherapy and eye enucleation, etc. Eye enucleation and radiotherapy have been the standard methods for controlling primary UM, but these measures lead to cosmetic defects and loss of vision [197]. Chemotherapeutic agents have been widely studied for treating liver metastases from uveal melanoma, but they do not prolong survival [198]. It is a challenge for drugs to reach the posterior segment of the eye, whether administered locally or systemically (Figure 8). In addition to being limited by physiological barriers such as the cornea, topical administration is also subject to specific protective mechanisms, such as blinking, lacrimation, and drainage, to facilitate rapid drug clearance. The systemic delivery of drugs to the eye is also tricky because the blood–retinal barrier acts like the blood–brain barrier, preventing the drug from diffusing from the blood to the retina. Due to all these mechanisms, the bioavailability of medicine in the eye is low, with an absorbed dose of less than 3% to 5% of the administered dose [199]. Recent advances in nanotechnology have provided new inspiration for drug delivery in UM and other ophthalmic diseases.

### 3.1. Nanoparticles for Drug Delivery

Topical ocular administration in the form of eye drops can effectively reduce systemic side effects, with high patient compliance due to the non-invasive nature of the procedure, which is of great importance. The complementary ocular delivery of drugs packaged in NPs has led to encouraging advances in the field of ocular drug delivery over the past decade. For example, dendrimers have been used for drug delivery to the anterior chamber [200]. Dendrimers can quickly enter or exit cells as they are 2–20 nm in size. Among the different NPs, lipid particles are suitable for ocular delivery due to their high biocompatibility [201] and their lipid components interacting with the outer lipid layer of tear fluid, thus promoting longer retention times [199]. In particular, solid lipid NPs and nanostructured lipid transporters represent promising strategies for ocular drug delivery due to their ability to incorporate a large number of drugs (up to 90%) and deliver them into the precorneal cavity and maintain a long residence time [199,201].

Based on experience with CM, various systemic treatments for UM have also been studied and clinical trials have demonstrated modest efficacy [202]. Sorafenib, a hydrophobic small molecule inhibitor of multiple tyrosine-protein kinases, can inhibit Raf kinases in the mitogen-activated protein kinase pathway, which is involved in most constituent activities of UM tumors [203]. For example, in previous studies, the experiments of Santonocito et al. demonstrated that a novel nanostructured microemulsion system with 0.3% sorafenib, administered as an ophthalmic formulation, could deliver an adequate amount of sorafenib to the retina, reducing pro-inflammatory and pro-angiogenic mediators [204]. In a recent study, Bonaccorso et al. investigated the ocular surface delivery of solid lipid NPs with sorafenib to treat UM and found that electrostatic interactions may provide enhanced adhesion, allowing the sustained release of the ocular surface of sorafenib [205]. Additionally, legumain, an aspartate endopeptidase, is highly expressed in choroidal melanoma [206]. With this, Luo et al. designed a chitosan-based nanocomposite with legumain-responsive properties that can be prepared rapidly. The nanomaterial exhibited increased cytotoxicity against a choroidal melanoma cell line and reduced cytotoxicity against normal human corneal epithelial cells, indicating the promising potential for the improved targeted delivery of chemotherapeutic drugs via the ocular surface [207].

Intravitreal injection is also an essential route for topical administration to the eye. It is widely used in treating various posterior segment diseases such as age-related macular degeneration and diabetic retinopathy. As an invasive procedure, it is crucial to increase locally injected drugs’ sustained and influential release. For example, Xie et al. developed an in situ biomimetic hydrogel system composed of natural biopolymers of collagen and HA. The set in situ gel delivery system gels in two minutes at 37 °C and exhibits excellent biocompatibility and slow degradation. The curcumin-loaded nanoparticle/hydrogel composite was able to release the payload for up to four weeks continuously. This new in situ nanoparticle/hydrogel composite has excellent potential for treating rare and devastating intraocular cancers [208]. The downregulation of hypoxia-inducible factor (HIF) has been shown to reduce the metastatic potential of various tumor cells, including breast, lung, and melanoma [209]. HIF-1α is a transcription factor that regulates the expression of secreted factors that mediate angiogenesis and tumor metastasis [210]. Xie et al. engineered a ternary chitosan/siRNA complex coated with HA as a molecular-based therapy targeting HIF-1α novel therapeutic strategy for UM. The complexes exhibited excellent cellular uptake and lysosomal escape capabilities with low cytotoxicity. Moreover, the ternary complex downregulates HIF-1α and VEGF expression in UM cells and successfully inhibits UM migration and invasion [211].

### 3.2. Nanoparticles for Radiotherapy

However, radiation therapy for small melanomas can control the tumor but may cause vision loss. Shields et al. investigated a new virus-like nanoparticle to treat small choroidal melanoma, which could induce tumor regression and minimize vision loss [212]. Moreover, since the absorbed energy doses of normal and tumor tissues are very similar, the maximum radiation dose has to be limited to the normal tissues surrounding the cancer. The use of radiosensitizers can solve this problem [213]. Nanoradiosensitizers are ideal candidates for enhancing tumor radiotherapy due to their high absorption from tumor tissues and their secondary electron productivity. For example, Ruan et al. showed that highly oxidized graphene quantum dots with good oxidative stress response and significantly high phototoxicity could be used as a novel nano-radiosensitizer for tumor radiotherapy [214]. Gold NPs have been used as radiosensitizers due to their high atomic number and photoelectric absorption coefficient [215]. Chang et al.’s study found that gold NPs can sensitize B16F10 melanoma cells to radiation and showed that NPs could accumulate in tumor cells [216]. Berbeco et al. found that even low concentrations of gold NPs produced vasculature-disrupting effects on tumor endothelial cells. Since NPs can both induce tumor cell apoptosis and disrupt their supporting vasculature when combined with radiation, they have the potential to support brachytherapy for the treatment of UM [217].

### 3.3. Nanoparticles for PDT and PTT

PDT and PTT therapy play an important role in UM. In recent years, NPs have been further studied to increase the therapeutic effect of PDT/PTT in UM. For example, Ahijado-Guzmán et al. demonstrated that UM cells could release and reuptake AuNS by studying photothermal effect changes between charged and uncharged UM cells [218]. Li et al. prepared nanocomposites based on fluorinated functionalized polysaccharides by anchoring perfluorocarbon and pyropheophorbide-a on the polymer chain of HA to supply oxygen to hypoxic areas. The photodynamic effect of this micelle was significantly enhanced due to the high oxygen affinity of the perfluorocarbon segment and the tumor targeting of HA [219]. Li et al. designed a microenvironment-triggered degradable hydrogel based on ultra-small (<5 nm) rare-earth NPs, which were encapsulated by dual-response hydrogels to release drugs in response to thermal energy and glutathione into the tumor microenvironment [220].

Additionally, it is generally believed that PDT can kill tumor cells through the production of ROS. It was initially thought that ROS was a cytotoxic by-product of tumor development and could be used to kill tumor cells [214]. However, nowadays, some studies have shown the opposite: ROS can promote tumorigenesis, malignant transformation, and chemoresistance [221]. Ding et al. investigated the effects of NPs (C-dot) on the metabolomics, growth, invasiveness, and tumorigenicity of UM cells. The results indicated that C-dots dose-dependently increased ROS levels in UM cells. At C-dot concentrations below 100 µg mL^−1^, C-dot-induced ROS promotes UM cell growth, invasiveness, and tumorigenicity; at 200 µg mL^−1^, UM cells undergo apoptosis [222]. The pro-cancer effect is mainly due to the C-dot-induced generation of ROS, the activation of Akt/mTOR signaling, and the increased metabolism of glutamine, thereby promoting UM cell proliferation and metastasis [222].

### 3.4. Nanoparticles for Imaging

Most patients with UM typically present with painless, nonspecific visual loss or visual changes (e.g., metamorphopsia, blurred vision, floaters, and so on) until the tumor is detected during a routine examination [223]. NPs and nanotechnology have great potential in the imaging of ocular diseases and UM for early detection and diagnosis. Gold nanocages synthesized by Raveendran et al. were first used successfully for high-contrast ocular imaging [224], with significant implications in UM. Additionally, iron oxide particles have been developed as a new MRI contrast agent for detecting UM in a rabbit model [225]. NPs promise to provide more precise and non-invasive procedures for diagnosing ocular melanoma.

## 4. Mucosal Melanoma

MM is a rare subtype of melanoma of unknown etiology. Primary MM can be found in the mucous membranes of the respiratory, gastrointestinal, and genitourinary tracts. About 1% of all melanomas are MM, usually arising from the mucosa of the head and neck, anorectal, or vulvovaginal mucosa [226]. Patients with MM are generally older, with a median age of 70 at diagnosis. Unlike CM, which is more common in men, MM is more common in women, with a female/male ratio of 1.85 to 1.0 [227]. The female predominance of MM may be due to the frequency of vulvovaginal melanoma, the most common subtype in women. In humans, the most common sites of MM are the head and neck [228]. MM is an aggressive malignancy with an inferior prognosis. In Europe, survival was calculated from 2277 European cases diagnosed between 2000 and 2007; survival rates at 1, 3, and 5 years were 63%, 30%, and 20% respectively [229]. While the incidence of CM has increased over time, the incidence of MM has remained stable [229]. Despite many technological advances in surgery, radiotherapy, and systemic therapy, the survival advantage of patients with MM has not increased [230].

Current treatments for MM are the same as for CM, although the biology is different. Extensive excision surgery is the treatment of choice. However, due to the anatomical limitations of MM and the multifocal growth of the lesion, wide local excision with negative margins is difficult to achieve, leading to high recurrence rates after surgical treatment [231]. Chemotherapy has similar effects in cutaneous and MM but does not significantly improve outcomes. Targeted therapy, in particular drugs targeting c-KIT, has been studied in patients with MM. Activating KIT mutations are relatively common in MM, found in approximately 40% of patients. However, in patients with KIT-mutated metastatic MM, c-KIT-targeting drugs failed to provide lasting responses [231]. As MM expresses lower levels of PD-L1 in the TME than CM [232], immunotherapy targeting PD-1 is ineffective. In conclusion, the treatment of MM still needs further research.

In terms of systemic therapy, nano-delivery systems could potentially facilitate the absorption and targeting of oral routes against the anatomical limitations of MM. A study reported PLGA NPs coated with chitosan (CS) and loaded with ferulic acid (FA). In simulated gastrointestinal fluid, approximately 15% of FA was released into gastric fluid, while minor release was observed in intestinal fluid. Moreover, the FA-loaded NPs showed the same efficacy as the free drug in cytotoxicity for melanoma cells. Therefore, CS-coated PLGA NPs might effectively treat oral gastrointestinal MM [233]. Moreover, another study showed that a nanoemulsion modified by tomato lectin could resist pepsin and trypsin digestion, and showed strong cellular immunity after targeting melanoma cells [234]. Moreover, curcumin has good antioxidant, anti-inflammatory, and antitumor activity. It has been reported that curcumin can inhibit the growth of melanoma cells by inducing a burst of ROS and destroying the mitochondrial membrane potential [235]. There are many studies on the treatment of CM. Due to curcumin’s low water solubility, it is difficult to mix with other ingredients, and its oral availability is low. Research by Yucel et al. has developed a submicrometric lipid-based carrier to encapsulate and deliver curcumin to intestinal epithelial cells [236]. Additionally, in a study by Pan et al., succinic anhydride-modified whey protein isolate and whey protein hydrolyzate were prepared and characterized as novel emulsifiers that help curcumin resist gastrointestinal digestion [237]. These studies provide a possibility for the local treatment of melanoma of the gastrointestinal mucosa by curcumin. These studies suggest a possible benefit for MM.

## 5. Discussion

With advances in research and clinical technology, survival expectations for patients with CM have improved significantly. For example, the median survival time increased from six to nine months in 2001 and 37 months in 2016. Nevertheless, problems remain. Melanoma is relatively difficult to administer given its rapid proliferation and early metastasis. In recent years, with the help of the design of different NPs, the problems encountered by drug delivery routes can be solved in a targeted manner. Some progress has been made, such as commercialized nanomedicines that use the passive targeted delivery of drugs or imaging agents. However, passive targeting does not fully achieve high drug enrichment at tumor sites. However, since melanoma does not have a highly specific high expression similar to CD19 in B-cell lymphoma and the commonly selected specific receptors are also partially distributed in other tissues, the selection of targeting ligands is still not highly convincing. For the same reason, chimeric antigen receptor T cell immunotherapy, which has become more prevalent in recent years, has some obstacles in its application to melanoma. New possibilities are offered through biofilms, such as melanoma cell membranes. In addition, due to melanoma pigment, the use of photothermal therapy is the suggested direction. At present, PDT and PTT have mature clinical applications, but the improvement of photosensitizers to further reduce the effects of adverse events and patient treatment efficacy have long-term significance. Currently, many studies use stimuli response and other methods to combine PDT/PTT and chemotherapy or immunotherapy. However, current research remains mainly at the level of in vitro experiments or animals, and further research is needed.

UM is the most common ocular tumor in adults. In addition to the inherent risks of life-threatening melanoma, UM typically results in visual impairment and loss in patients. The rapid growth rate of the tumor always causes visual occlusion, and the radical treatment during the treatment process usually results in an impossible visual for the patient. It is crucial to use NPs to target tumors for preserving the patient’s vision as much as possible.

Besides, in addition to complex treatment, CM is difficult to diagnose. Early detection, diagnosis, treatment, and prognosis are essential factors in the treatment of tumors. As a rare disease, there are few studies on mucosal melanoma, and most focus on clinical observation and clinical research. Some experiments provide the possibility of gastrointestinal drug treatment. However, further research is needed to improve diagnostic sensitivity to diagnose mucosal melanoma.

## 6. Conclusions

Melanoma is a relatively rare tumor caused by the malignant transformation of melanocytes in various organs, including the skin, mucous membranes, eye area, etc. It is also a very aggressive tumor, prone to metastasis [238]. Patients with metastatic melanoma often respond poorly to conventional chemotherapy, with median overall survival, disease-specific survival, and recurrence-free survival of 37, 45, and 48 months [239]. Therefore, the prompt and effective treatment of melanoma is very crucial. The use of nanoplatforms may overcome some of the limitations of current melanoma treatments. Nanotechnology is a multidisciplinary field that aims to revolutionize the detection and treatment of cancer by engineering biomaterials such as molecules, atoms, and supramolecules in the nanometric range (1 to 100 nm) [16]. NPs possess unique properties, such as permeability, hydrophilicity, stability, porosity, and large specific surface area. These allow NPs to facilitate drug delivery and the entry of minor compounds into cancer cells and improve the intratumoral concentration of drugs, reducing the effect on healthy tissues [175]. In this review, we summarize the limitations of current treatments for CM, UM, and MM and the auxiliary role of NPs in different treatment modalities. Currently, nanoparticle delivery methods cannot provide conclusions for all the challenges that we face in drug delivery. New delivery systems still need to be created. Our goal is to make an effective, non-invasive way to deliver drugs to patients so as to improve treatment outcomes. We believe that in the future, nanomaterials can be used to advance the combination treatment and early diagnosis of melanoma and effectively improve the prognosis of melanoma.

## Figures and Tables

**Figure 1 pharmaceutics-14-02090-f001:**
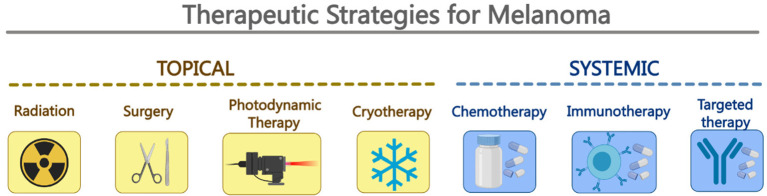
Therapeutic strategies for melanoma (this figure was created with MedPeer http://image.medpeer.cn/).

**Figure 2 pharmaceutics-14-02090-f002:**
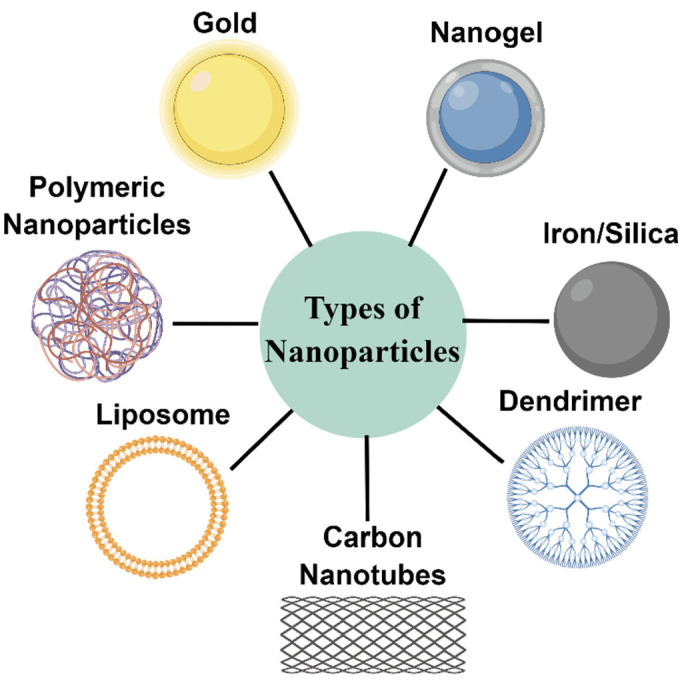
Types of nanoparticles (this figure was created by Figdraw).

**Figure 3 pharmaceutics-14-02090-f003:**
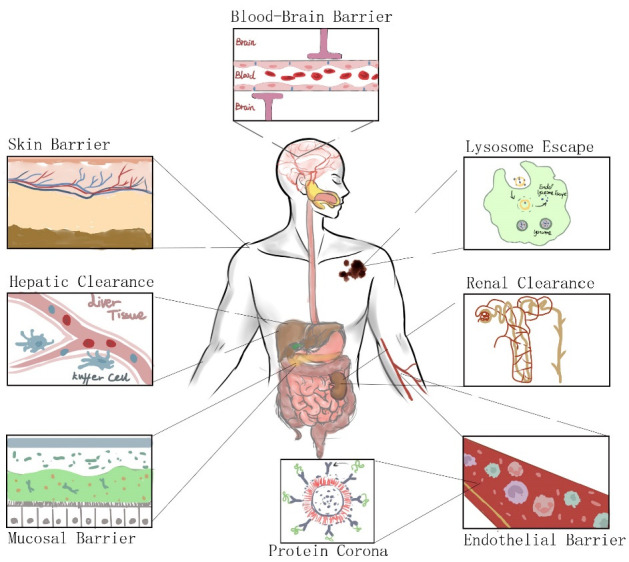
Physiological barriers to systemic and topical drug therapy.

**Figure 4 pharmaceutics-14-02090-f004:**
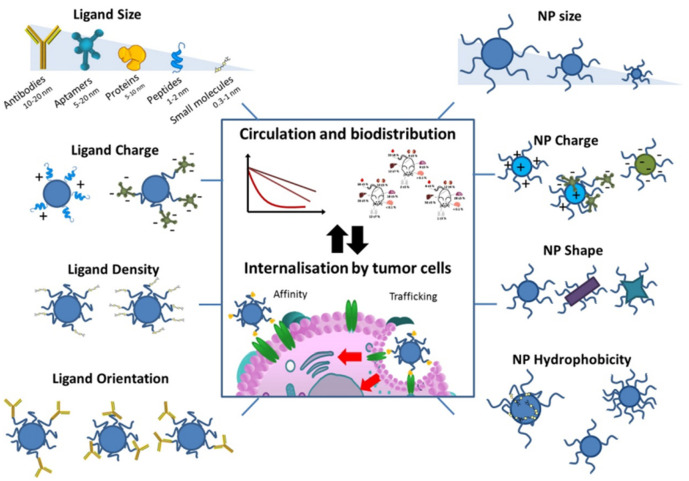
The physicochemical properties of the ligand and the NP affect their blood circulation profiles, their biodistribution, and their ability to be internalized by cancer cells (Reprinted from [74], with permission from Elsevier).

**Figure 5 pharmaceutics-14-02090-f005:**
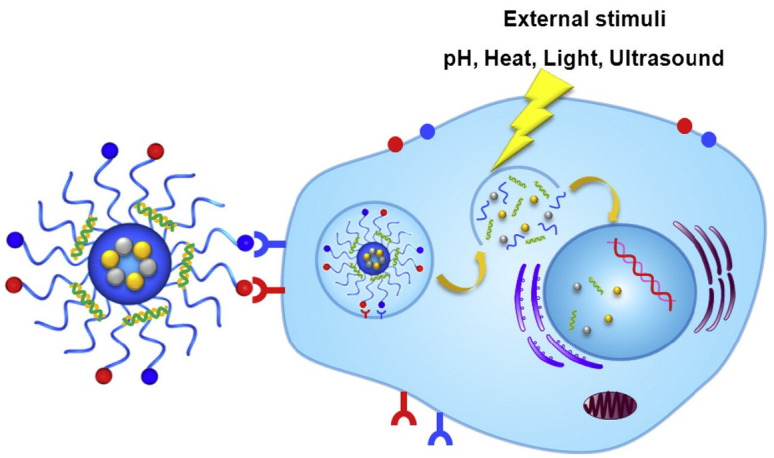
Schematic illustration of the stimuli-triggered drug release from the NPs (Reprinted from [23], with permission from Elsevier).

**Figure 6 pharmaceutics-14-02090-f006:**
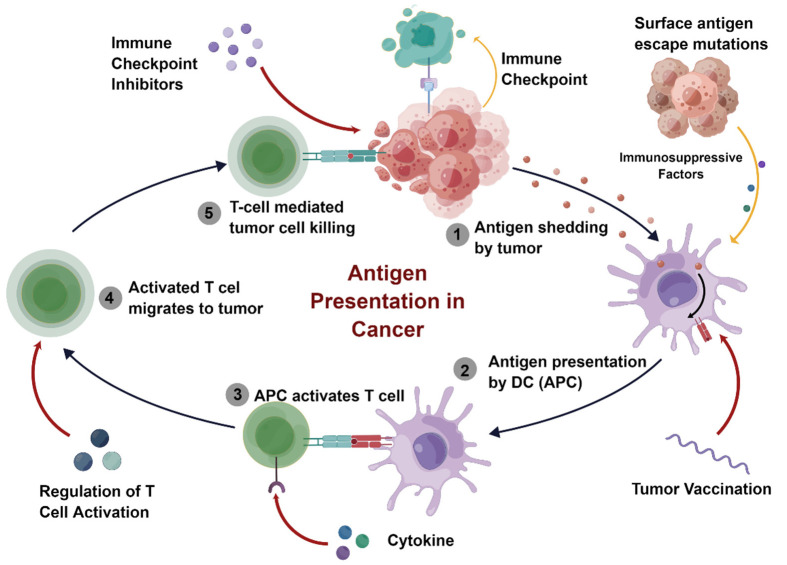
The cancer antigen-presenting machinery and the involvement of nanoparticles in the antitumor immunity pathway (this figure was created By Figdraw). Dendritic cells capture antigens to present them to T cells, leading to T-cell activation. Activated T cells target and kill tumor cells. Tumor tissue can affect DC presentation through the secretion of immunosuppressive factors and the role of immune checkpoints in suppressing T cells. Nanoparticles can promote DC and T-cell activation by delivering tumor vaccines, cytokines, etc., which can promote the presentation of tumor antigens. At the same time, the antitumor effect of T cells can be enhanced by delivering immune checkpoint inhibitors.

**Figure 7 pharmaceutics-14-02090-f007:**
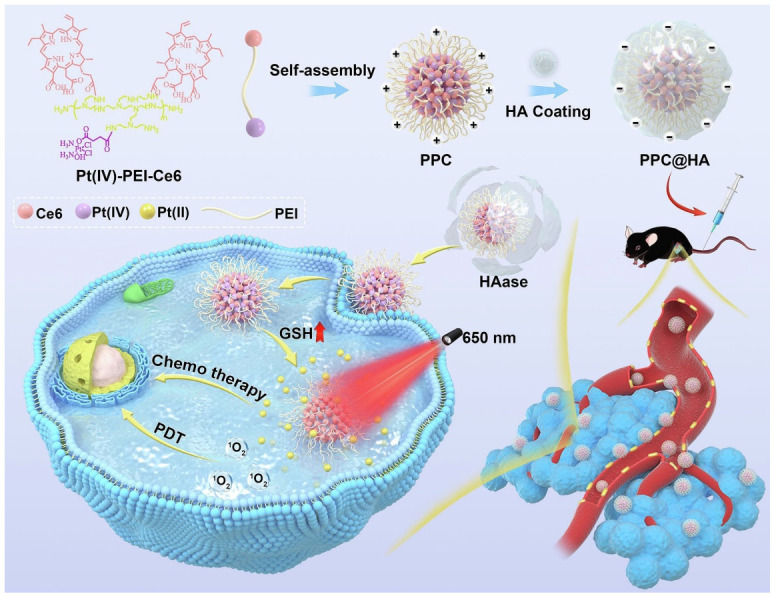
Synthetic procedure of charge-reversal micelles (PPC@HA) and schematic illustration of the targeted drug delivery and enhanced cellular uptake of the designed chemo-photodynamic nanomedicine for anti-melanoma treatment. The HA coating increases the targeted delivery of PPC@HA into the tumor tissue, and hyaluronidase in the tumor tissue induces HA degradation to form positively charged PPC, resulting in enhanced cellular uptake by B16 cells. Finally, PPC@HA exhibits improved biodistribution in the tumor tissue and significantly inhibits tumor growth through chemo-photodynamic combination therapy (Reprinted from [181], with permission from Elsevier).

**Figure 8 pharmaceutics-14-02090-f008:**
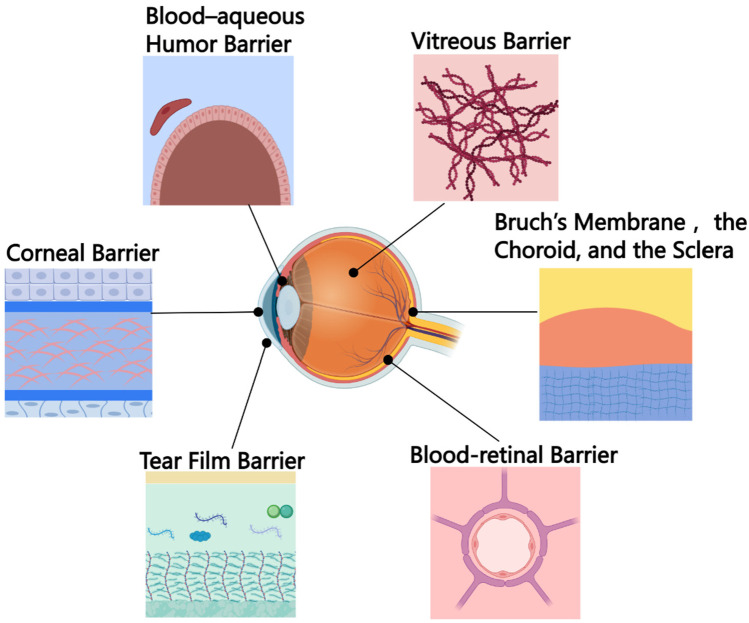
Barriers to drug delivery to the posterior segment of the eye (This figure was created with MedPeer http://image.medpeer.cn/).

## Data Availability

Not applicable.

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
