# Peer review of "Advances in the Application of Nanomaterials to the Treatment of Melanoma"

_pharmaceutics, 2022, doi:10.3390/pharmaceutics14102090_

Round 1
Reviewer 1 Report
Manuscript titled " Advances in the Application of Nanomaterials to the Treatment of Melanoma" is a very interesting paper in the field of nanomaterials for cancer treatment. Overall structure is of good quality, paragraphs are clear and cover an appropriate range of application in nanotachnology for cancer treatment. However, authors should improve the manuscript in several parts:
1. Firstly, authors should describe the potential oral availability of curcuminoids for chemopreventive and anticancer therapies ( you can cite 10.1016/j.ijpharm.2015.08.039 )
2. In discussion authors should describe the oil/water nano-emulsion loaded with cobalt ferrite oxide for cancer diagnosis and therapy and preclinical evidences actually available ( you can cite 10.1016/j.nano.2016.08.022)
3. In introduction or discussion, authors should describe the potential benefits derived from liposomes or exosomes for delivert of nutraceuticals in cancer like ascorbic acid or resveratrol or curcuminoids. Authors should add a proper description of this point and the potential benefits of ascorbic acid-loaded nanomedicines for melanoma therapy ( cite 10.3390/antiox9121182 )
Author Response
Dear reviewer:
We greatly appreciate the reviewer’s constructive suggestions on our manuscript. We have carefully revised our paper according to the comments, and the major revised portions were highlighted in yellow. Here are the point-by-point responses to the comments as listed below, along with a clear indication of the location of the revision:
1.Firstly, authors should describe the potential oral availability of curcuminoids for chemopreventive and anticancer therapies ( you can cite 10.1016/j.ijpharm.2015.08.039 )
--We very much appreciate this comment as it points us to the studies we overlooked. We have read and summarized the relevant literature and included them succinctly in our manuscript (Line 1180-1189 mark in yellow).
2.In discussion authors should describe the oil/water nano-emulsion loaded with cobalt ferrite oxide for cancer diagnosis and therapy and preclinical evidences actually available ( you can cite 10.1016/j.nano.2016.08.022)
--Thank you for the suggestion, we have included it succinctly in our manuscript (Line 250 mark in yellow).
3.In introduction or discussion, authors should describe the potential benefits derived from liposomes or exosomes for delivert of nutraceuticals in cancer like ascorbic acid or resveratrol or curcuminoids. Authors should add a proper description of this point and the potential benefits of ascorbic acid-loaded nanomedicines for melanoma therapy ( cite 10.3390/antiox9121182 )
--Thank you for the suggestion, we have added them succinctly in our manuscript (Line 424 and 277 mark in yellow).
Reviewer 2 Report
The manuscript entitled: "Advances in the Application of Nanomaterials to the Treatment of Melanoma" from Wang et al, in my opinion, in its present form, is not recommended for publication in Pharmaceutics, requiring a major review. The manuscript does not have enough novelty, since other recent review articles contain similar information (e.g. doi: 10.3390/ijms22041873, 10.3390/molecules26040785). However, these papers are not divided in cutaneous, uveal and mucosal melanoma, which can be an opportunity for the current manuscript.
Additional reasons for drawing this conclusion are as follows:
1) The English must be reanalysed in whole document and several corrections throughout the text have to be performed. Some examples:
- Both British and American English are used (e.g. “tumour” and “tumor”)
- “to treat melanoma treatment” (line 87)
- “PEG-PLGA block co-polymers 240 has long been used” (line 240)
- “This biodegradable polymers” (line 251)
2) Many errors of abbreviations are found in the manuscript, making it frustrating for the reader. Some examples:
- Several abbreviations defined several times (e.g. MPS – lines 154, 157, 231)
- Abbreviations not defined (e.g. NC, TME)
- Several defined abbreviations that do not appear again in the text (e.g. EGCG, CAF, PPTB)
- Several abbreviations are defined, but continue to appear without being abbreviated (e.g. NP)
3) The manuscript is highly descriptive. I suggest to include, for example, a section with a “critical overview”. Probably this alteration could differentiate this manuscript from other similar published papers.
4) I propose to include the BBB (blood-brain barrier) in Figure 3, since metastatic melanoma often spreads to brain.
5) Repetition of the sentence in lines 194-199.
6) Line 320 – “better effect” than….
7) In active targeting, it was not mentioned, for example, ligands for transferrin and folic acid receptors, which are overexpressed on the surface of melanoma cells
8) Cytokines are endogenous immune mediators. I propose “cytokine-based drugs” instead of “cytokines” (line 585)
9) Line 710 – One advantage of PDT is patient tolerance? Please, reformulate the sentence.
10) I do not understand the section “3.5. Conclusions”. Why only uveal melanoma has a different conclusion?! Please, include that information in the overall conclusion, which should be more critical than the current conclusion at section 5.
Author Response
Dear reviewer:
Thank you for your decision and constructive comments on my manuscript. We have carefully considered the suggestion of Reviewer and make some changes. We have tried our best to improve and made some changes in the manuscript.
The green part that has been revised according to your comments. Revision notes, point-to-point, are given as follows:
1) The English must be reanalysed in whole document and several corrections throughout the text have to be performed. Some examples:
--We are very sorry to have made these errors in the manuscript. We have already correcting the language of this article, both what you mentioned and what you did‘t mention, thank you for your detailed suggestions.
2) Many errors of abbreviations are found in the manuscript, making it frustrating for the reader. Some examples:
--We are very sorry for our previous abbreviation errors. We have revised all parts involving abbreviations according to the rules. Thank you for your suggestion.
3) The manuscript is highly descriptive. I suggest to include, for example, a section with a “critical overview”. Probably this alteration could differentiate this manuscript from other similar published papers.
-- We have added a "Discussion" chapter in the penultimate part to discuss and evaluate the current application of nanoparticles in melanoma, thank you for your suggestion!
4) I propose to include the BBB (blood-brain barrier) in Figure 3, since metastatic melanoma often spreads to brain.
-- Thank you for your careful review and suggestions, we have revised this figure.
5) Repetition of the sentence in lines 194-199.
-- Thank you for your careful review and suggestions, we have revised this sentence.
6) Line 320 – “better effect” than….
-- Thank you for your careful review and suggestions, we have revised this.
7) In active targeting, it was not mentioned, for example, ligands for transferrin and folic acid receptors, which are overexpressed on the surface of melanoma cells
--We very much appreciate this comment as it points us to the studies we overlooked. We have read and summarized the relevant literature and included them succinctly in our manuscript (lines 417-444 mark in green).
8) Cytokines are endogenous immune mediators. I propose “cytokine-based drugs” instead of “cytokines” (line 585)
-- Thank you for your careful review and suggestions, we have revised this.
9) Line 710 – One advantage of PDT is patient tolerance? Please, reformulate the sentence.
-- Thank you for your careful review and suggestions, we appreciate we have revised this sentence.(line 879)
10) I do not understand the section “3.5. Conclusions”. Why only uveal melanoma has a different conclusion?! Please, include that information in the overall conclusion, which should be more critical than the current conclusion at section 5.
-- Thank you for your careful review and suggestions, we have deleted this part,and add this to “conclusion”.
Reviewer 3 Report
The paper entitled "Advances in the Application of Nanomaterials to the Treatment of Melanoma" presents a review regarding the use of nanoparticles and their limitations for treatment of different types of melanoma.
The paper presents a thorough overview for the use of nanoparticles in treatment of cutaneous, uveal and mucosal melanoma and it is appropriate to be published in Pharmaceutics with following minor observations:
Rows 156, 181, correct NCs to NPs.
2. Row 200, give details about the type of nanoparticles used in ref 35.
3. Row 205, the same details about type of NPs would be useful to be specified.
4. Rows 335, idem.
5. Row 444, enzyme mimetics such as…
6. Rows 578, 579, correct met-al, nanopar-ticles
7. Rows 705-706, finish sentence: Since photosensitizers tend to bind preferentially to low density lipoproteins (LDL)…
Author Response
Dear reviewer:
Thank you for your decision and constructive comments on my manuscript. We have carefully considered the suggestion of Reviewer and make some changes. We have tried our best to improve and made some changes in the manuscript.
Revision notes, point-to-point, are given as follows:
1. Rows 156, 181, correct NCs to NPs.
Thank you for your careful review and suggestions, we have revised this part.
2. Row 200, give details about the type of nanoparticles used in ref 35.
Thank you for your careful review and suggestions, we have revised this part.(line 227)
3. Row 205, the same details about type of NPs would be useful to be specified.
Thank you for your careful review and suggestions, we have revised this part.(line 232)
4. Rows 335, idem.
Thank you for your suggestions, since the original article is the breast cancer targeting design, the purpose of modification of NPs is to target HER2, which is not suitable for melanoma. Therefore the details about the type of nanoparticles did not be mentioned, and it is only listed as evidence of the actively targeted NPs.
5. Row 444, enzyme mimetics such as…
Thank you for your careful review and suggestions, we have revised this part.(line 557)
6. Rows 578, 579, correct met-al, nanopar-ticles
Thank you for your careful review and suggestions, we have corrected this.
7. Rows 705-706, finish sentence: Since photosensitizers tend to bind preferentially to low density lipoproteins (LDL)…
Thank you for your suggestions, we have already finished this sentence.
Reviewer 4 Report
In this manuscript, the authors propose a very detailed review of the studies and use of nanomaterials to the treatment of melanoma.
Many reviews on similar concern are already available in the literature such as: Li J. et al, Nanomedicine, 2015, 769; Mundra V. Nanomedicine, 2015, 10, 2613; Pautu V et al., Pharmacol Res, 2017, 126, 31 and a very recent one of Chen H et al, Frontiers in Oncology, 2022 (12) 928797. Nevertheless, regular updates and a variety of reports appear to be of specific interest in a such highly evolving field.
The review appears to be well written and covers a large area of studies and concerns. However, several points appear to be noteworthy and problematic.
The authors state (line 286): “Dacarbazine is the only chemotherapeutic drug for melanoma approved by the USFDA”. Such statement is particularly strange and troublesome. What about the different BRAF and MEK inhibitors such as vemurafenif, dabrafenib, encorafenib, tametinib or cobimetinib ? Furthermore no study such as the ones of Fu Y. et al. in Exp. Cell Research, 2020, 396, 112275; Tham H.P. et al., ACS Nano, 2018, 12, 11936 or Nguyen H.T et al. J. Controlled Release, 2021, 329, 524 (those papers are only examples) on nanoformulations of such compounds are reported and discussed in the review. It might be a choice of the authors not to incorporate them in their review but such choice needs to be explained and discussed. If not, a part of the review devoted to such studies on nanomaterials using targeted therapy APIs must be added and discussed.
In their conclusion, the authors state “the overall median survival is only 6-9 months”, and refer for such statement to a very old and totally outdated paper of 2001. Since then, many new treatments have hopefully been available to the patients and the survival rate increased. Updated references and data are mandatory.
Author Response
Dear reviewer:
We greatly appreciate the reviewer’s constructive suggestions on our manuscript. We have carefully revised our paper according to the comments, and the major revised portions were highlighted in blue. Here are the point-by-point responses to the comments as listed below, along with a clear indication of the location of the revision:
(1) About the statement that“Dacarbazine is the only chemotherapeutic drug for melanoma approved by the USFDA”: We discussed Dacarbazine in detail in our manuscript because it is a chemotherapeutic drug that has been used earlier and widely studied. But we are very sorry for our inappropriate expression and we have rephrased this statement regarding your suggestion (Line *347 marked in blue).
(2)关于上述文献:我们非常感谢这一评论,因为它指出了我们忽略的研究。我们阅读并总结了相关文献,并简明扼要地将其包含在我们的手稿中(标有蓝色的529-535行)。
(3)关于整体中位生存率:我们为引用过时的结论深表歉意。我们再次检索了关于黑色素瘤的流行病学文献,根据最近的文献,我们可以发现黑色素瘤的中位总生存期,疾病特异性生存期和无复发生存期分别为37个月,45个月和48个月。我们已将上述内容及其参考文献添加到我们的手稿中,并将它们标记为蓝色(第1231-1234行标记为蓝色)。
Round 2
Reviewer 2 Report
The authors have answered to my remarks and amended, wherever necessary, the text of the manuscript accordingly. At present, I can recommend acceptance of the manuscript for publication.
Author Response
We appreciate the reviewer for the affirmation of our revised manuscript, and we also thank the Reviewer for his/her comments which definitely improved the quality of our study.
Reviewer 4 Report
The authors have taken into consideration the remarks. To be noted that a large part of their response was onl written in chinese and had to be translated in english to get an appropriate understanding of the input in the revised version.
All remarks have been taken into consideration in the revised version.
Additional comments:
- On line 534, the term "drugs" is far too vague and large. It might be more precise to mention an indication such as "MAPK inhibitors". Nevertheless such paragraph which deals with studies on NP incorparation of highly clinically used and beneficial drugs, remains particularly short in comparison to others which are focused on much less used drugs.
- on line 1199, the sentence "These factors lead to difficulties in the treatment of melanoma, incredibly advanced treatment" can not be understood and must be revised.
Author Response
Dear reviewer:
We greatly appreciate the reviewer’s constructive suggestions on our manuscript. We have carefully revised our paper according to the comments, and the major revised portions were highlighted in blue. Here are the point-by-point responses to the comments as listed below, along with a clear indication of the location of the revision:
About the statement that“- On line 534, the term "drugs" is far too vague and large. It might be more precise to mention an indication such as "MAPK inhibitors". Nevertheless such paragraph which deals with studies on NP incorparation of highly clinically used and beneficial drugs, remains particularly short in comparison to others which are focused on much less used drugs.”:
--We are sorry for missing some very important relevant references and did not describe this part clearly in the last version of the manuscript. We have further refined the summary of this part of the literature and summarized the relevant literature and included them succinctly in our manuscript (lines 529-553 marked in blue).
About the the sentence "These factors lead to difficulties in the treatment of melanoma, incredibly advanced treatment" can not be understood and must be revised:
--Thank you for your careful review and suggestions. We are very sorry this sentence was difficult to read. We have revised this sentence.(Line 1216 marked in blue).
Moreover, we're very sorry that part of the last response was written in Chinese because of the plugin.We carelessly missed the mistake.The original text of the last response in English is as follows:
(1) About the statement that“Dacarbazine is the only chemotherapeutic drug for melanoma approved by the USFDA”: We discussed Dacarbazine in detail in our manuscript because it is a chemotherapeutic drug that has been used earlier and widely studied. But we are very sorry for our inappropriate expression and we have rephrased this statement regarding your suggestion (Line *347 marked in blue).
(2) About the mentioned literature: We very much appreciate this comment as it points us to the studies we overlooked. We have read and summarized the relevant literature and included them succinctly in our manuscript (Line 529-535 marked in blue).
(3) About the overall median survival: We deeply apologize for quoting an outdated conclusion. We searched the epidemiological literature on melanoma again and based on the most recent literature we could find the median overall survival, disease-specific survival and no relapse survival of melanoma were 37, 45 and 48 months, respectively. We have added the above and its references to our manuscript and marked them in blue (Line 1231-1234 marked in blue).
Thank you for your careful reading and patience.